# Contemplative Practices Behavior Is Positively Associated with Well-Being in Three Global Multi-Regional Stanford WELL for Life Cohorts

**DOI:** 10.3390/ijerph192013485

**Published:** 2022-10-18

**Authors:** Tia Rich, Benjamin W. Chrisinger, Rajani Kaimal, Sandra J. Winter, Haley Hedlin, Yan Min, Xueyin Zhao, Shankuan Zhu, San-Lin You, Chien-An Sun, Jaw-Town Lin, Ann W. Hsing, Catherine Heaney

**Affiliations:** 1Stanford Prevention Research Center, Department of Medicine, Stanford School of Medicine, Stanford University, Stanford, CA 94035, USA; 2Department of Social Policy and Intervention, University of Oxford, Oxford OX1 2ER, UK; 3Penumbra, Inc., Alameda, CA 94502, USA; 4Quantitative Sciences Unit, Department of Medicine, Stanford University, Stanford, CA 94035, USA; 5Department of Epidemiology and Population Health, Stanford School of Medicine, Stanford University, Stanford, CA 94305, USA; 6Chronic Disease Research Institute, The Children’s Hospital, and National Clinical Research Center for Child Health, School of Public Health, School of Medicine, Zhejiang University, Hangzhou 310058, China; 7Department of Nutrition and Food Hygiene, School of Public Health, School of Medicine, Zhejiang University, Hangzhou 310058, China; 8School of Medicine, Data Science Center, College of Medicine Fu-Jen Catholic University, New Taipei City 24205, Taiwan; 9Department of Gastroenterology and Hepatology, E-Da Hospital, Kaohsiung City 82445, Taiwan; 10Stanford Cancer Institute, Stanford School of Medicine, Stanford University, Stanford, CA 94035, USA; 11Department of Psychology, Stanford University, Stanford, CA 94305, USA

**Keywords:** contemplative practices, health promotion, mindfulness, meditation, well-being, WELL for life

## Abstract

Positive associations between well-being and a single contemplative practice (e.g., mindfulness meditation) are well documented, yet prior work may have underestimated the strength of the association by omitting consideration of multiple and/or alternative contemplative practices. Moreover, little is known about how contemplative practice behavior (CPB) impacts different dimensions of well-being. This study investigates the relationship of CPB, consisting of four discrete practices (embodied somatic-observing, non-reactive mindfulness, self-compassion, and compassion for others), with multiple dimensions of well-being. As with other canonical lifestyle behaviors, multiple contemplative practices can be integrated into one’s daily routine. Thus, it is critical to holistically consider these behaviors, extending them beyond a simple uni-dimensional measure (e.g., daily mindfulness meditation practice). We developed an integrative measure of four types of contemplative practice and found it to be significantly associated with a multi-dimensional measure of well-being. Importantly, our findings were from three large global multi-regional cohorts and compared against better-understood lifestyle behaviors (physical activity). Data were drawn from California/San Francisco Bay Area, (*n* = 6442), Hangzhou City (*n* = 10,268), and New Taipei City (*n* = 3033). In all three cohorts, we found statistically significant (*p* < 0.05) positive associations between CPB and well-being, both overall and with all of the constituent domains of well-being, comparable to or stronger than the relationship with physical activity across most well-being outcomes. These findings provide robust and cross-cultural evidence for a positive association between CPB and well-being, illuminate dimensions of well-being that could be most influenced by CPB, and suggest CPB may be useful to include as part of fundamental lifestyle recommendations for health and well-being.

## 1. Introduction

It is well-established that multiple health behaviors compound and intersect over the course of an individual’s day. For instance, overall daily diet is comprised of different foods consumed as meals and snacks throughout the day, and physical activity is the summation of different forms of activities, including moderate (e.g., housework, gardening, walking to work) and vigorous (e.g., lifting weights, jogging) forms. Similarly, the construct of allostatic load has been developed to emphasize the deleterious effects of cumulative physiological “wear and tear”, rather than singular stressors or behaviors [1]. In each of these constructs, the sum of behaviors may be far more critical for health outcomes and overall well-being than the individual behaviors alone.

Contemplative practices include a set of activities that quiet the striving mind, cultivate awareness, develop conscious attention modulation capabilities, promote presence, connect the individual to something larger than their own life, and develop and sustain an experience of being known/seen, safe, soothed, and secure [2]. These practices deepen and expand awareness and discernment by cultivating the capacity to bear witness to lived experience—internally, relationally, and collectively. Furthermore, the strengthening of awareness and discernment by contemplative practice facilitates the expansion of healthy engagement with greater complexity in one’s individual life and the lives of others. Positive associations between well-being and a single contemplative practice (e.g., mindfulness or compassion) are well documented [3,4,5], but the association with combined multiple contemplative practice behavior (CPB) is less understood. In this study, we hypothesize that more frequent CPB, including, but not limited to, mindfulness meditation, may be associated with greater well-being. This paper simultaneously evaluates multiple aspects of CPB, yielding a more comprehensive measure of individuals’ CPB practice overall. The study also assesses multiple dimensions of well-being, providing a unique opportunity to determine specific elements (domains) of well-being that are most likely to be impacted by CPB. Information from a specific domain of well-being and its relationship with specific CPB will help inform the design of effective targeted intervention studies in the future to promote well-being in individuals and in communities.

### 1.1. Introducing Contemplative Practices

There is a growing interest in contemplative practices among professionals in public health, community mental health, wellness, and medicine in addition to the enduring interest among traditional spiritual religious and/or psychological practitioners. In 2021 Davidson called for a broader approach to the study of contemplative practices. He stated “I will conclude with a plea that mindfulness be situated within a more expansive framework to cultivate well-being and that interventions be appropriately broadened to include additional elements that are necessary for human flourishing. The cultivation of well-being will be framed as an urgent public health need and strategies to disseminate practices at scale require investigation” [6]. This study is a response to that call.

Davidson and Dahl define contemplative practices as “practical methods to bring about a state of enduring well-being or inner flourishing,” and include physical and mental behaviors that are thought to affect a variety of psychological constructs [7]. Contemplative practices emphasize self-awareness, self-regulation, and/or self-inquiry to enact a process of well-being, which may include psychological and/or spiritual transformation, and/or self-transcendence [7,8,9,10,11,12]. In addition to fostering states that promote individual well-being, CPB enhances traits that may also contribute to social welfare through prosociality, equanimity, altruism, compassion, and ethics [7,13,14].

Contemplative practices include tools and techniques from the world’s traditions of spirituality and religion, and indigenous systems of healing and health promotion. Thus, most contemplative practices originated as part of integrated coherent lifestyle systems intended to strengthen an individual’s ability to *thrive*, *create* innovations that address the needs of humanity and society, and *serve* the health and well-being of all of life [15]. The integrated systems provided philosophical and theoretical frameworks that have examined and offered explanations for the evolution and expression of the natural interplay of the mental, emotional, and spiritual facets of human life that support biopsychosocialspiritual development, health, and well-being [16,17,18]. These systems include but are not limited to the Taoist Five Element System, the Eight Limb teachings of Raja/Ashtanga Yoga, the Buddhist Eightfold Path, the Islamic Path of Dhikr practice, and the Christian path delineated in *The Cloud of Unknowing* [19], a spiritual guide to contemplative prayer that contributed to the development of Centering Prayer.

Unifying these culturally diverse traditions and systems is the principle that contemplation offers a sense of connection with the source of all of life and the direct experience of awe and feelings of reverence and gratitude. Furthermore, each system offers several contemplative practices that facilitate an increased frequency, duration, and depth of contemplation throughout daily life, not only during formal contemplative practice, thus giving meaning to every moment.

Today there are opportunities to learn a variety of contemplative practices from the diverse world traditions and systems. However, the modern dissemination of contemplative practices has frequently occurred in dispersed fragments rather than through the transfer of an entire tradition or system. Furthermore, the traditional systems were not developed within the context of modern life and all of its complexities. Instead, most of the systems emerged related to monastic life in agrarian societies. Ken Wilber’s Integral Life Practice [18] has offered a framework for the modern day. Nevertheless, there remains a need for further research to construct a cohesive comprehensive understanding of the means by which to best incorporate contemplative practices into modern day life. Our study aims to contribute to the further development of the theory and framework for the modern-day application of contemplation and contemplative practices. This study builds upon and expands beyond the evidence on “a la carte” contemplative practices (e.g., mindfulness meditation, compassion cultivation, or hatha yoga, etc., considered independently).

### 1.2. Contemplative Practices and Well-Being

It is well documented that a variety of different contemplative practices are related to an array of positive biopsychosocial outcomes, including support for connections between mindfulness meditation and immune system biomarkers [20]), and an association of self-reported mindfulness meditation practice with physical activity, with meditators less likely to be inactive, and more likely to meet guidelines for optimal physical activity [21,22,23]. A recent review of workplace-based mindfulness programs suggested that such efforts may help improve multiple dimensions of psychological functioning among employees [24]. In addition, a meta-analysis found that compassion-based interventions can improve self-reported psychosocial and interpersonal outcomes [25]. Furthermore, Western psychological interventions that incorporate classic Buddhist contemplative practices have been shown to promote a sense of purpose and meaning, thus fostering more enduring contentment [26].

Many studies of CPB have focused exclusively on mindfulness meditation, producing evidence of positive benefits [27,28]. In 2021, Davidson noted “MBIs are truly a model of a transdiagnostic intervention that may potentially have beneficial impact across a wide range of conditions and populations” [6]. Studies have found that even a short amount of meditation practice can reduce rumination and trait anxiety, increase empathy and self-compassion [29,30], develop healthy distress tolerance, beneficial emotional regulation and emotional stability [28,31,32], increase happiness [7], and foster conscious regulation of mental attention. MacCoon and colleagues found that in a randomized control trial, compared to a validated active control group [33], meditation-naive participants in an 8-week meditation intervention experienced decreased reactivity to affective stimuli and enhanced automatic emotion regulation [34]. Importantly, these benefits were further enhanced among participants who were long term meditation practitioners.

Prolonged contemplative practice training may have the potential to impact not only perceptions of well-being but also biological processes underlying health status. In a meditation study, participants in a three-month retreat program had both positive biological and psychological effects. Compared to the wait-list controls who were matched for age, body mass index, and prior meditation experience, individuals in the meditation retreat program had improved telomerase activity and immune cell functioning as well as decreased neuroticism and increased purpose in life [35].

Among studies that have investigated the impact of CPB among clinical populations, there is some evidence for both physical and psychological benefits. Research on contemplative practices in patients with cardiac disease has shown “encouraging results” for improving perceived physical and mental quality of life, as well as systolic and diastolic blood pressure [36]. Among cancer patients, a mindfulness-based stress reduction program, which included meditation and yoga training as well as interpersonal discussion exercises, was found to improve patients’ quality of life and decrease negative experiences of stress, as well as lower cortisol levels, systolic blood pressure, and pro-inflammatory cytokines [37,38]. A meta-analysis by Hoffman and colleagues found moderate support for mindfulness-based therapies’ effectiveness for reducing anxiety and improving mood among clinical populations [39]. Other reviews have found preliminary evidence to support mindfulness interventions to treat pain, depression, and addiction [40].

### 1.3. Understanding Multiple Practices: Contemplative Practice Behavior

Investigating the typology and combined effect of multiple contemplative practices on well-being will improve our insight into how CPB affects well-being and health. Most studies on contemplative practices to date have assessed the relationship between a single contemplative practice, such as hatha yoga, mindfulness meditation, or compassion cultivation, and a specific health or well-being outcome.

To fill this gap, our study uses a summary index measure of contemplative practices and a multi-dimensional measure of well-being to assess the individual and combined contributions of CPB on both overall well-being and on nine domains of well-being, leveraging survey responses from a total of 19,743 individuals from three global study sites. We used a set of four contemplative practices, including embodied observing meditation, non-reactive meditation, self-compassion, and compassion for others to measure CPB. We used the WELL for Life survey [41] to measure multi-dimensions of well-being to determine the associations between CPB and overall well-being and its nine constituents.

Prior research into the benefits of CPB has generally taken a reductionistic approach by focusing on one specific contemplative practice, such as mindfulness meditation, and by evaluating the impact on a relatively narrow range of well-being dimensions. The present research aims to extend this work by examining the association of a more expansive measure CPB with a broader range of well-being dimensions (e.g., financial well-being, creativity, and spirituality). By incorporating an inclusive definition of CPB in our assessment we were able to implement the same survey across three different cultural contexts in which engagement in specific CPB may vary to examine the robustness of the associations of CPB with nine specific dimensions of well-being.

## 2. Materials and Methods

### 2.1. Study Setting and Design

Begun in 2015, the Well for Life Study aims to quantify and contextualize individual well-being, investigate patterns and determinants of well-being in a large, multi-ethnic, and global multi-regional population, and to promote well-being in individuals and communities [41]. Participants provided informed consent and the study was approved by the Stanford University Institutional Review Board.

Participants were primarily recruited from three regions: the San Francisco Bay Area region of California, New Taipei City, and Hangzhou City in Zhejiang Province. In the US and in New Taipei City, participants were recruited via community partners, community outreach events, mailing lists, and social media. In the US, recruitment strategies also led to responses from outside the Bay Area and California, though 69.3% of the sample was from the Bay Area and a further 6.9% from California; thus, we subsequently use the label “CA/Bay Area”. In Hangzhou City, participants were recruited using stratified quota sampling from three of the city’s nine districts [42].

Data were collected via online questionnaires in the CA/Bay Area, or by self-administered surveys during a visit to a university lab in Hangzhou and New Taipei City. The questionnaire gathered information about participant demographics, medical history, contemplative practices, and well-being. All questionnaire items were presented in English (CA/Bay Area cohort) or in Mandarin Chinese (Hangzhou and New Taipei City cohorts) translated from the English version. In order to accommodate cultural differences (detailed below), some of the demographic items varied slightly across the cohorts. 

### 2.2. Independent Variable

*Contemplative Practice Behavior (CBP).* Contemplative practice behavior (CPB) encompasses four distinct practices, each measured by a single item to reduce the participant burden. Our four CBP items reflected behaviors that cultivate each of the dimensions of contemplative practice included in the S-ART model forth by Vago and Silbersweig Self-Awareness, -Regulation, and -Transcendence, defined as a positive relationship between self and other that transcends self-focused needs and increases prosocial characteristics, such as compassion [11].

Our four items were based on the factors identified in prior research to be most representative of the salient processes associated with the benefits of contemplative practice [43,44,45,46,47]. The practice of embodied somatic self-awareness was measured by the frequency of *embodied-observing practices* (i.e., pausing routine activities for at least five minutes for breathing deeply, gently stretching, noticing your senses). The practice of mindfulness and self-regulation was measured by *non-reactive practices* (i.e., pausing routine activities for at least five minutes for observing emotions and thoughts as they arise rather than being caught up in them). Compassion practice was measured by the frequency of *self-compassion practice* (i.e., pausing routine activities for at least five minutes to observe and modify the way one is thinking to offer more compassion, love, or kindness to oneself) and *compassion practice toward others* (i.e., pausing routine activities for at least five minutes to observe and modify the way one is thinking to offer more compassion, love, or kindness toward others). The frequency of each contemplative practice was measured on a five-point scale (0–4; Never, Almost never, Sometimes, Fairly often, Very often). An overall CPB score was calculated as the sum of the four practice items.

Gu et al.’s [43] assessment of the Five Factor Mindfulness Questionnaire [48,49] identified that the following four facets load best into one score: Describing, Acting with Awareness, Nonjudging of Inner Experience, and Nonreactivity to Inner Experience function as four subscales, while the Observing factor functions as a separate measure. Observing refers to attending or noticing internal and external experiences (e.g., sounds, emotions, thoughts, bodily sensations, smells). Furthermore, Nonreactivity to Inner Experience has been identified as a significant component of the mechanism by which contemplative practices are beneficial [3,32,50,51,52,53,54,55]. Thus, we measured behaviors of embodied observing and behaviors of non-reactivity to inner awareness. Similarly, the self-compassion and compassion behaviors were measured because they have been identified to be dimensions of contemplative practice that contribute to benefits through mechanisms distinct from behaviors focused on embodied observing and non-reactivity to inner experiences [56,57,58,59].

### 2.3. Outcome Measure (Well-Being and Its Nine Domains)

Well-being was assessed using the 53-item WELL survey, the development of which has been described previously [41,60]. Briefly, the WELL survey was developed by the Stanford Well for Life Study through grounded theory and qualitative research that identified domains of well-being in various cultural groups to create a tool for understanding well-being that is valid across cultures. Standard questions in each domain from internationally validated surveys were used to construct the WELL survey in nine domains of well-being. In New Taipei City, a 52-item survey was administered, without the single-item self-rated health question used in the CA/Bay Area and Hangzhou cohorts. Formative assessment of the survey included cognitive interviews as recommended by Willis and colleagues [61]. Table 1 shows the list of the nine domains, associated definitions, and sample items.

For each of the nine well-being domains, a score from 0 to 10 was created based on the responses to the constituent items: 14 for stress and resilience, 13 items for social connectedness, 11 for experience of emotions, 5 for sense of self, 4 for physical health, 2 for purpose and meaning, 1 for financial security and satisfaction, 1 for spirituality and religiosity, and 1 for exploration and creativity. Higher scores on each domain indicate more optimal levels of well-being. For example, a higher score for the experience of emotions domain indicates more frequent positive emotions and less frequent negative emotions. The domain scores were summed to create the overall well-being score (WELL score). Each of the nine domains were scored 0–10, and an unweighted overall well-being score was calculated by summing each of the domain scores. For ease of interpretation, the score was re-scaled to 100.

### 2.4. Test–Retest Reliability and Convergent Validity

A sub-sample of initial survey participants in the US were invited to participate in a re-administration of the questionnaire one week later. The test–retest correlation for the WELL score was 0.92 (*n* = 92). Moreover, as part of the test–retest administration, participants were asked to complete the WHO-5 [62] in order to assess the association of the WELL score with the well-validated WHO-5. The correlation of the WELL score with the WHO-5 was 0.73. Results of a confirmatory factor analysis for the US WELL score had good model fit with rmsea = 0.059 and cfi = 0.852. Cronbach alphas for the domains that were measured with multi-item scales were: Resilience 0.92, Stress 0.78, Social connectedness 0.89, Negative emotions 0.85, Positive emotions 0.86, Sense of self 0.87, Purpose and Meaning 0.86, and Physical health 0.76.

### 2.5. Covariates

#### 2.5.1. Physical Activity

Given the robust and long-standing evidence base for physical activity’s (PA) benefits to overall health status [63] and the evidence suggesting a positive relationship of PA with perceived physical health, quality of life, and well-being [64,65,66], PA was included as a covariate in all analyses. In the CA/Bay Area and New Taipei City, PA was measured using the Stanford Leisure-Time Activity Categorical Item (L-Cat 2.2) [67]. This is a single item measure that asks people to read through six descriptions of activity levels and choose the one that best describes their level of activity during the last month. Responses from 1 (“I did not do much physical activity […]”) to 6 (“Almost daily, that is, five or more times a week, I did vigorous activities such as running or riding hard on a bike for 30 min or more each time”). The full text of these survey questions is available in Appendix A. In Hangzhou, PA was measured with the International Physical Activity Questionnaire (IPAQ). This measure includes a number of questions regarding PA (e.g., time spent walking) and total time of moderate and vigorous activity every day in the past week. The PA score was generated by adding the times and weighting them based on activity to arrive at a categorical variable that could be classified as light, moderate or vigorous activity [68].

#### 2.5.2. Demographic Characteristics

The WELL survey also included questions on age, gender, marital status, employment status, educational attainment, and ethnicity and race (in the Bay Area only). Several of these variables were measured in slightly different ways between cohorts, which are noted in Table 1.

### 2.6. Statistical Analysis

The distribution of each of the four CPB items was assessed as well as the Spearman correlation (pairwise) of items with one another. Continuous variables were centered at their median values, and binary variables were coded as −0.5 and 0.5 so that estimates represent averages. Dummy variables were created for each of the categorical variables and were coded as 1–1/m and −1/m, where m is the number of categories in each variable.

The association of CPB with the WELL score and the nine domain-specific scores were modeled separately for each study site (CA/Bay Area, Hangzhou, and New Taipei City), using hierarchical multivariate linear regression. For each outcome, a series of models were fitted: Model 1 (demographic covariates alone), Model 2 (Model 1 covariates and PA), and Model 3 (Model 2 covariates and CPB). Multivariate Wald tests were used to sequentially compare each model (e.g., Model 1 vs. Model 2, Model 2 vs. Model 3) and to determine the significance of the additional covariates [69]. Models were adjusted for gender, age (continuous), education (high school or less, some college or associate degree, bachelor’s degree, and post-graduates), marital status (married or cohabitating, single and other) and work status (working, students, retired, not working). In the Bay Area cohort, race (white/Caucasian, Asian or Pacific Islander, Black/African American, and Multiracial/other race) and ethnicity (Hispanic or not) covariates were also included; race/ethnicity variables were not surveyed in Hangzhou or New Taipei City as these populations are mostly Han Chinese (95% in New Taipei City and 98% in Hangzhou). For the Hangzhou cohort, the education categories were recategorized as high school or less, some college education, and college degree or above. The largest category for each variable served as the reference group in each cohort (see Table 2).

Missing values of all covariates and dependent variables were imputed via multiple imputation [70,71] using the R package [72] (R Core Team 2016), mice4 version 2.46.0 [73]. Ten iterations of imputation were carried out using predictive mean matching, logistic regression, and polytomous regression imputation for continuous, binary, and categorical data, respectively; summaries of the imputed data are presented alongside the original datasets as Appendix A. Mean estimates, 95% confidence intervals, and adjusted R^2^ values were calculated from the pooled regression estimates for all models. All analyses were performed in R version 3.3.3.

## 3. Results

### 3.1. Demographic Characteristics

Table 2 provides a descriptive summary of participant demographic characteristics by study site. Overall, 19,743 participants were included in this study from three separate cohorts: 6442 from the CA/Bay Area, 10,268 from Hangzhou, and 3033 from New Taipei City. All three cohorts had high proportions of female participants (between 60 and 71%). Compared to the Hangzhou and New Taipei City cohorts, the CA/Bay Area cohort was younger (mean age of 41.4 years versus 54.4 and 53.2, respectively), more highly educated (68.7% having a bachelor’s degree or higher), and mostly employed (67.6% versus 29.6% in Hangzhou and 53.6% in New Taipei City). The CA/Bay Area cohort also included a larger proportion of single individuals (42.8%), whereas among the Hangzhou and New Taipei City cohorts, most participants reported their status as married or cohabiting (85.2% and 75.6%, respectively).

### 3.2. Descriptive Statistics for CPB, PA, and Well-Being

Unadjusted overall scores of well-being were significantly (*p* < 0.001) higher among CA/Bay Area participants (mean = 59.1, SD = 12.0) compared to the other two cohorts, which had similar means (Hangzhou: mean = 55.9, SD = 9.2; New Taipei City: mean = 55.3, SD = 11.0). The highest average domain-specific score was financial security and satisfaction for the CA/Bay Area, social connectedness for Hangzhou, and sense of self for New Taipei City; the lowest average domain-specific score was spirituality and religiosity for all cohorts. The four CPB items were highly correlated in all three cohorts, with pairwise Spearman correlations ranging from 0.42 to 0.72 in the CA/Bay Area, 0.36 to 0.62 in Hangzhou, and 0.43 to 0.73 in New Taipei City. Average CPB was highest for the New Taipei City cohort (9.20, SD = 3.21), where the most frequent practice reported was embodied mindfulness (see Table 3). In the CA/Bay Area cohort, average CPB was 8.20 (SD = 3.72), and compassion toward others was the most frequent practice. The Hangzhou cohort had a roughly similar CPB mean (8.90, SD = 2.95), with compassion toward others as the most frequent practice. Physical activity scores (measured using L-Cat 2.2) were significantly (*p* < 0.001) lower among the New Taipei City cohort (2.60, SD = 1.23) compared to the CA/Bay Area (3.50, SD = 1.45). Among the Hangzhou cohort, 49.4% of participants reported recent physical activities that were classified as “vigorous” (measured using IPAQ). Table 3 describes these health behaviors and well-being outcomes across the three cohorts.

### 3.3. Contemplative Practice Behavior and Well-Being

Across all three cohorts, CPB was significantly (*p* < 0.001) associated with well-being (see Figure 1, Table 4). With every standard deviation increase in CPB, the overall WELL score increased by 1.22 points (SE = 0.03, 95% CI = 1.15–1.28) for the CA/Bay Area cohort, 1.16 (SE = 0.03, CI = 1.10–1.22) for the Hangzhou cohort, and 1.73 (SE = 0.05, CI = 1.63–1.83) for the New Taipei City cohort. The addition of CPB to overall WELL score models produced a notable effect in all three cohorts: adjusted R^2^ increased in Models 2 to 3 by 0.16 to 0.30 in the CA/Bay Area, from 0.04 to 0.17 in Hangzhou, and 0.12 to 0.36 in New Taipei City (see Table 4).

All of the WELL domains were significantly (*p* < 0.05) and positively associated with CPB score in each of the three cohorts. Figure 2 illustrates the coefficient estimates and confidence intervals for associations between CPB, PA, and the nine well-being domains among the three cohorts. The relative contribution of the CBP variable, as measured by each model’s adjusted R2 value, varied widely between domains (see Table 5); however, across the three cohorts, the domains of purpose and meaning, exploration and creativity, and spirituality and religiosity were most sensitive to the addition of the CPB variable.

As expected, physical activity, which was entered as a covariate, was also positively and significantly (*p* < 0.001) associated with overall well-being among the CA/Bay Area, Hangzhou, and New Taipei City cohorts (Table 4). Figure 1 illustrates the coefficients of association and 95% confidence intervals for PA and CPB in fully adjusted models of overall well-being. Significant positive associations were found for all well-being domains in the New Taipei City and CA/Bay Area cohorts, except for spirituality and religiosity, where no significant relationship was observed for the New Taipei City cohort, and a significant negative association was found for the CA/Bay Area cohort. Among the Hangzhou cohort, PA was significantly and positively associated with six domains of well-being. Figure 2 illustrates these associations across the three cohorts and nine domains of well-being.

## 4. Discussion

We found significant associations between a summary index of four distinct contemplative practices (CPB) and multi-dimensional well-being (WELL score) in a large cohort of individuals from three different global regions. The unique finding of the association between a combination of four CPBs with a multidimensional assessment of well-being across three cohorts from different global regions provides evidence that the effect of contemplative practice behaviors on well-being transcends regions and cultures. The magnitude of the positive associations was larger for the New Taipei City cohort, but findings were similar for the CA/Bay Area and Hangzhou cohorts, suggesting that culture may play a role in the size of the association of CPB with well-being.

The significant and positive associations between CPB and most WELL domains in all three cohorts suggest the importance of CPB for psychosocial and mental health and other health outcomes. Of the nine domains in WELL, six assess aspects of mental and psychosocial health: experience of emotions, exploration and creativity, purpose and meaning, sense of self, social connectedness, and perceived stress and resilience. Thus, positive associations with these domains echo the results from previous studies on contemplative practices and psychosocial outcomes [7,74,75]. Positive associations also emerged between CPB and three domains of well-being not traditionally included in measures of physical or mental health, including financial security and satisfaction (Hangzhou and New Taipei City), and physical health (CA/Bay Area and New Taipei City). These correlations suggest an association of CPB with outcomes across a wide range of factors related to well-being beyond those that are typically included in studies of mental and physical health.

In the context of previous research including both observational and experimental studies that has clearly documented a positive relationship between PA and well-being [65,66,76,77], we sought to examine the association between CPB and well-being over and above the contribution of PA to well-being. Our data show that that both PA and CPB were independently associated with well-being and its constituent domains. While a direct quantitative comparison of the regression coefficients for CPB when predicting well-being and those for PA when predicting well-being was not appropriate given the different levels of measurement of these two concepts in this study, our findings do suggest that the associations between CPB and well-being followed a comparable positive pattern to those between PA and well-being. Notably, the magnitude of these associations varied between domains. For instance, the coefficient of association for CPB was greater than that of PA within the purpose and meaning domain across all three cohorts.

Further research is needed to generate a clearly defined recommendation for the frequency and intensity of contemplative practice behaviors, similar to the recommendations for performing 10,000 steps for physical activity [78], sleeping for 7–9 h per night, and eating five servings of fruits and vegetables per day [79]). Until new studies have illuminated and clarified optimal contemplative practice behavioral recommendations, this study suggests that as with physical activity, sleep, and fruits and vegetables, “some contemplative practice is better than none”.

As evidence for the health-enhancing potential of CPB accumulates, government-sponsored surveillance systems have an opportunity to build assessments of CPB into their data collection and agenda-setting strategies, which may inspire greater attention to this aspect of lifestyle across the public and private sectors. Inclusion of contemplative practice behaviors as part of the fundamental lifestyle recommendations for health and well-being will likely lead to an increase in interventions and curricula that focus on contemplative practices behaviors in health promotion programs in schools as well as in public health and health care systems. Many current health promotion programs’ budgets mainly focus on physical activity and nutrition, and may offer access to gyms, pools, nutritious meals, and farmer’s markets, as well as fitness and nutrition assessments. However, there seems to be relatively fewer resources dedicated to contemplative spaces and contemplative practice behavior assessments and skill-building opportunities. For example, while the *Healthy People 2020* materials included the newly added section “Health-related Quality of Life and Well-being” that measures components of well-being comparable to those found to be positively associated with CPB in this study, it does not specify an assessment of contemplative practices behaviors [80]. Similarly, the 2019 Center for Disease Control’s (CDC) Behavioral Risk Factor Survey did not ask about CPB [81]. The CDC’s redesigned National Health Interview Survey (NHIS) Sample Adult Questionnaire pain management assessment includes contemplative practices options (yoga or tai chi, meditation, guided imagery, or other relaxation techniques); however, it does not assess contemplative practices as one of the annually measured core health-related behaviors [82]. Nevertheless, an expert panel on community health promotion convened by the CDC suggested that more integrated and inclusive approaches to well-being were needed, including changes to research and funding priorities [83]

While this study speaks to the broader possible benefits of CPB, other research [15] suggests that health and well-being promotion interventions that cultivate and support contemplative practice behavior are feasible, affordable, and adaptable. Broader approaches such as these could be fundamental to achieving the broad visions set forth by national and international frameworks, such as *Heathy People 2030* and the World Health Organization constitution, which states: “Health is a state of complete physical, mental and social well-being and not merely the absence of disease or infirmity” [84].

### Limitations

Several limitations should be noted. First, the cross-sectional design precluded a causal understanding of the relationships identified in this study; future longitudinal designs and mediation analyses may help unpack any causal mechanisms at play in these relationships. Second, qualitative research methods may be better suited to illuminating the specific ways in which contemplative practices contribute to well-being for research participants. Third, when the single-item domains (three of nine) are used as the dependent variables in certain regression models, the residuals are not normally distributed; the relatively large size of the datasets employed in the analyses partially mitigated this issue [85]. Fourth, among the three cohorts, age and education distributions differed significantly, and could potentially have influenced the results within each cohort and the comparisons across cohorts. However, the diversity across sites did seem to provide some internal replications to support the robustness of the findings, which followed a generally consistent pattern between cohorts. Finally, biometric data were not included in this analysis, although the use of such measures, particularly those identified in prior research on the health benefits of contemplative practices, such as markers of immune system function [20,86], salivary cortisol, heart rate, heart rate variability [87], blood pressure [36], electroencephalogram (EEG) [88], and MRI and fMRI [89], would strengthen future investigations.

## 5. Conclusions

As with other canonical lifestyle behaviors, multiple contemplative practices can be integrated into one’s daily routine. Thus, it is critical to holistically consider these behaviors, extending them beyond a simple uni-dimensional measure (e.g., minutes of daily mindfulness mediation practice). We developed an integrative measure of four types of contemplative practice and found it to be significantly associated with a multi-dimensional measure of well-being. Importantly, our findings were from three large global multi-regional cohorts and compared against better-understood lifestyle behaviors (physical activity), broadening their applicability to settings around the globe.

## Figures and Tables

**Figure 1 ijerph-19-13485-f001:**
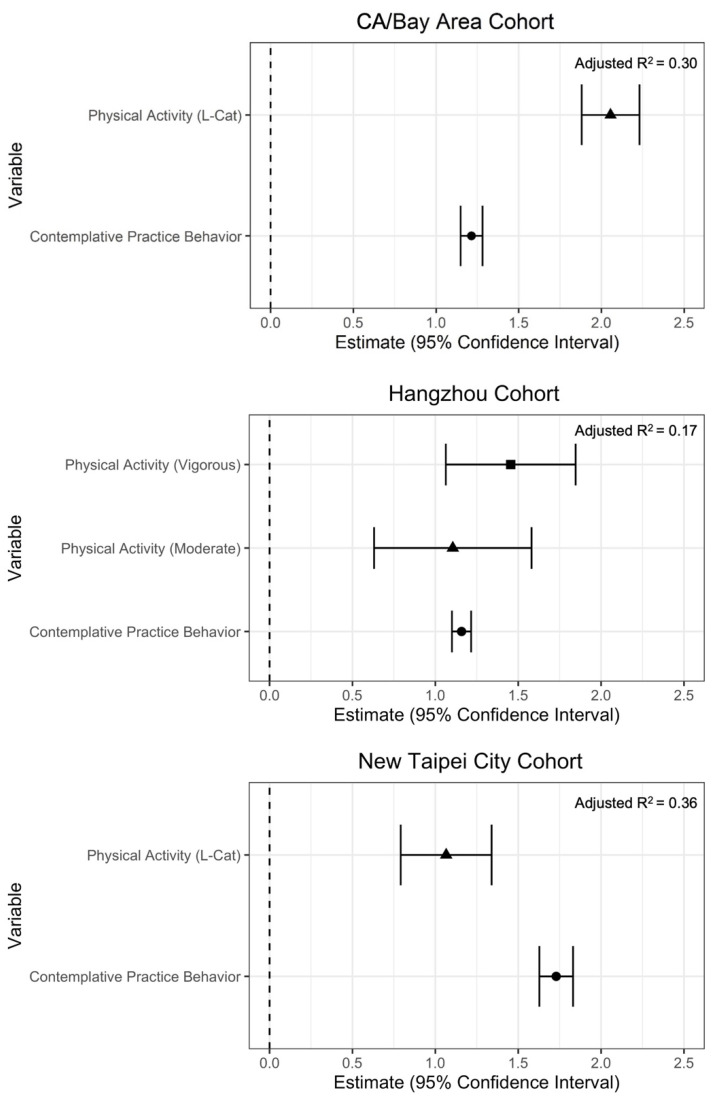
Associations between overall WELL scores, contemplative practice behavior, and physical activity. Note: A different physical activity measure was used for the Hangzhou cohort (International Physical Activity Questionnaire; categorical variable with PA-Low as reference group) than for the CA/Bay Area and New Taipei City cohorts (L-Cat 2.2; ordinal variable, 1–6).

**Figure 2 ijerph-19-13485-f002:**
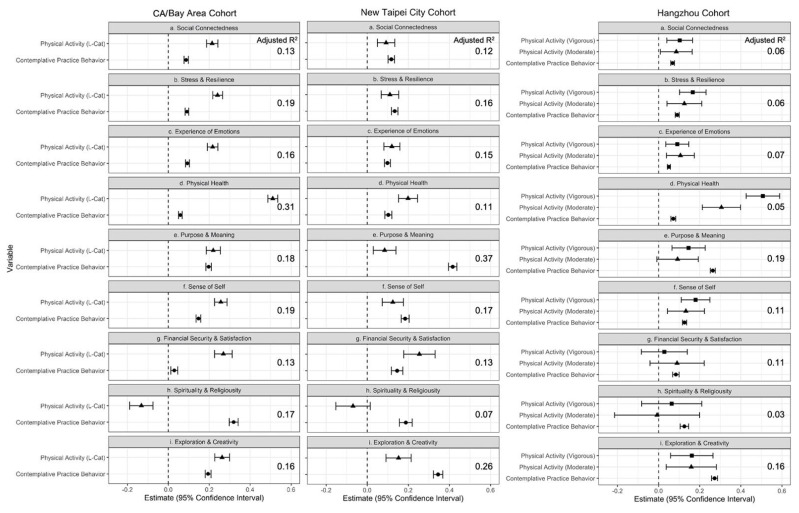
Associations between Contemplative Practice Behavior, Physical Activity, and Well-Being Domains. Note: A different physical activity measure was used for the Hangzhou cohort (International Physical Activity Questionnaire; categorical variable with PA-Low as reference group) than for the CA/Bay Area and New Taipei City cohorts (L-Cat 2.2; ordinal variable, 1–6).

**Table 1 ijerph-19-13485-t001:** Stanford WELL for Life: Constituent Domains of Well-Being.

Domain	Definition	Example Items	Number of Items
Social Connectedness	Positive or negative relationships with others and how they influence your well-being.	During the last two weeks, how often did you feel … …that you lacked companionship? …that there were people you could talk to? … that you were a part of a group of friends?	13
Stress and Resilience	Stress: Feelings of overload and an inability to balance or manage tasks Resilience: Ability to adapt to change and bounce back after hardship.	During the last two weeks, how often have you felt that you were not able to give enough time to the important things in your life? How confident are you that you can bounce back quickly after hard times?	14
Experience of Emotions	How often you experience both pleasant and unpleasant emotions.	During the last two weeks, how often did you feel … calm? … drained?	11
Physical Health	Perception of your own health status, i.e., energy levels, ability to resist illness, physical fitness, and experience of pain.	Compared to others of your own age, how would you rate your health? During the last two weeks, how often did your energy level allow you to do the things you WANT to do, as opposed to only the things you have to do?	4
Purpose and Meaning	Having a sense that aspects of your life provide purpose and meaning, i.e., goals, dreams, and being part of something larger than yourself.	How often does your daily life include experiences that give your life … purpose? … meaning?	2
Sense of Self	The extent to which you feel you know yourself, can express your true self, have self-confidence, and feel good about who you are.	During the last two weeks, how often did you feel … accepting of yourself? … that you were interested in your daily activities?	5
Financial Security and Satisfaction	Your perception of having enough money to meet your needs.	During the last year, how often have you had enough money to meet your needs?	11
Spirituality and Religiosity	The extent to which spiritual and religious beliefs, practices, communities, and traditions are important in your life.	How important are spiritual or religious beliefs in your day-to-day life?	1
Exploration and Creativity	Having opportunities to grow as a person and to explore new experiences and ways of thinking.	How often do you engage with opportunities to challenge yourself and grow … as a person?	1

**Table 2 ijerph-19-13485-t002:** Demographic Characteristics of Study Participants ^1^.

	CA/Bay Area ^2^ (*n* = 6442)	Hangzhou (*n* = 10,268)	New Taipei City (*n* = 3033)
**Age, mean (SD)**	41.4 (17.2)	53.2 (14.1)	54.4 (11.5)
**Gender ^3^**			
Female	4586 (71.2)	6187 (60.3)	2064 (68.1)
Male	1754 (27.2)	4081 (39.7)	969 (31.9)
*Missing*	28 (0.4)	0 (0.0)	0 (0.0)
**Educational attainment**			
High school or less	819 (12.7)	7499 (73.0)	1593 (52.5)
Some college	1221 (19.0)	1255 (12.2)	
Bachelor’s degree ^4^	2084 (32.4)	1208 (11.8)	1173 (38.7)
Post-graduate/professional degree	2273 (35.3)	NA	264 (8.7)
*Missing*	45 (0.7)	306 (3.0)	3 (0.1)
**Employment status**			
Working	4356 (67.6)	3036 (29.6)	1627 (53.6)
Not Working	539 (8.4)	2171 (21.1)	507 (16.7)
Retired	472 (7.3)	4723 (46.0)	835 (27.5)
Student	1047 (16.3)	32 (0.3)	13 (0.4)
*Missing*	28 (0.4)	306 (3.0)	51 (1.7)
**Marital status**			
Married or cohabiting	2758 (42.8)	8748 (85.2)	2293 (75.6)
Single	2606 (40.5)	555 (5.4)	407 (13.4)
Other	1047 (16.3)	659 (6.4)	329 (10.8)
*Missing*	31 (0.5)	306 (3.0)	4 (0.1)

^1^ Figures reported are from pre-imputation datasets. ^2^ Race/ethnicity data were collected for the Bay Area cohort and appear in Appendix A. ^3^ Transgender participants (*n* = 74, 1.1%) were also recorded in the CA/Bay Area WELL survey.^4^ For the Hangzhou cohort, the highest option for educational attainment was “college and above”, and is presented here under bachelor’s degree.

**Table 3 ijerph-19-13485-t003:** Health Behaviors and Well-Being Outcomes across the Study Sample ^1^.

	CA/Bay Area (*n* = 6442)				Hangzhou (*n* = 10,268)				New Taipei City (*n* = 3033)			
Variables	Mean	sd	*n* Missing	% missing	Mean	sd	*n* Missing	% Missing	Mean	sd	*n* Missing	% Missing
**Contemplative Practice Behavior (CPB)**	8.20	3.72	100	1.55	8.90	2.95	361	3.50	9.20	3.21	3	0.10
Embodied Mindfulness	2.20	1.17	68	1.06	2.30	1.02	360	3.50	2.60	0.94	1	0.03
Non-Reactive Mindfulness	1.80	1.21	70	1.09	1.90	0.99	361	3.50	2.10	1.05	0	0.00
Self-Compassion	1.90	1.08	66	1.02	2.30	0.90	360	3.50	2.20	0.98	1	0.03
Compassion toward Others	2.40	1.06	69	1.07	2.30	0.91	360	3.50	2.30	0.93	1	0.03
Physical activity (L-Cat)	3.50	1.45	92	1.43					2.60	1.23	60	1.98
Physical activity (IPAQ) ^2^					6.10	4.26	1201	11.70				
WELL overall score	59.10	11.96	147	2.28	55.90	9.16	360	3.50	55.30	11.03	10	0.33
**Domain-specific scores** Experience of emotions	5.90	1.58	36	0.56	6.70	1.11	354	3.40	6.50	1.35	5	0.17
Exploration and creativity	6.90	2.23	71	1.10	5.00	2.29	360	3.50	5.30	2.32	1	0.03
Financial security and satisfaction	7.70	2.62	75	1.16	7.00	2.39	360	3.50	6.40	2.61	0	0.00
Physical health	6.80	1.61	27	0.42	6.20	1.55	332	3.20	6.10	1.58	0	0.00
Purpose and meaning	6.80	2.15	88	1.37	6.30	1.81	360	3.50	5.90	2.23	1	0.03
Spirituality and religiosity	4.70	3.58	57	0.88	3.70	3.04	358	3.50	4.70	2.82	0	0.00
Social connectedness	6.70	1.67	41	0.64	7.30	1.21	354	3.40	7.00	1.44	1	0.03
Stress and resilience	6.30	1.50	35	0.54	6.60	1.34	350	3.40	6.30	1.49	5	0.17
Sense of self	7.30	1.89	51	0.79	7.20	1.51	358	3.50	7.20	1.84	1	0.03

^1^ Numbers reported are from pre-imputation datasets and are unadjusted for demographic covariates; ^2^ A different physical activity measure was used for the Hangzhou cohort (International Physical Activity Questionnaire) than for the CA/Bay Area and New Taipei City cohorts (L-Cat 2.2, score: 1–6).

**Table 4 ijerph-19-13485-t004:** Regression coefficient estimates for CPB and PA in the models for the overall WELL score.

	Cohort	Estimate	Std. Error	Lower CI	Upper CI
Contemplative Practice Behavior	CA/Bay Area	1.22	0.03	1.15	1.28
	New Taipei City	1.73	0.05	1.63	1.83
	Hangzhou	1.16	0.03	1.10	1.22
Physical Activity ^1^	CA/Bay Area	2.06	0.09	1.88	2.23
	New Taipei City	1.07	0.14	0.79	1.34
	Hangzhou Vigorous	1.45	0.20	1.06	1.85
	Hangzhou Moderate	1.11	0.24	0.63	1.58

^1^ A different physical activity measure was used for the Hangzhou cohort (International Physical Activity Questionnaire) than for the SF Bay Area and New Taipei City cohorts (L-Cat 2.2), and Light physical activity was used as the reference group.

**Table 5 ijerph-19-13485-t005:** Adjusted R^2^ Values with Confidence Intervals and Wald Tests of Significance for the Models in Hierarchical Regressions.

	Model 1	Model 2	Model 3
Domain	Demographic Covariates	Demographic Covariates + PA ^1^	Demographic Covariates + PA ^1^ + CPB
**CA/Bay Area**	*Adj R* ^2^ *(95% CI)*	*Adj R* ^2^ *(95% CI), p-value*	*Adj R* ^2^ *(95% CI), p-value*
WELL overall score	0.09 (0.08–0.1)	0.16 (0.15–0.18), <0.001	0.30 (0.29–0.32), <0.001
Experience of Emotions	0.07 (0.06–0.08)	0.11 (0.1–0.13), <0.001	0.16 (0.14–0.18), <0.001
Exploration and Creativity	0.03 (0.02–0.03)	0.06 (0.05–0.08), <0.001	0.16 (0.15–0.18), <0.001
Financial Sec. and Satisfaction	0.11 (0.09–0.12)	0.13 (0.11–0.14), <0.001	0.13 (0.11–0.15), <0.001
Physical Health	0.08 (0.07–0.09)	0.29 (0.27–0.31), <0.001	0.31 (0.29–0.33), <0.001
Purpose and Meaning	0.04 (0.03–0.05)	0.07 (0.06–0.08), <0.001	0.18 (0.16–0.2), <0.001
Sense of Self	0.06 (0.05–0.07)	0.11 (0.09–0.12), <0.001	0.19 (0.17–0.21), <0.001
Social Connectedness	0.06 (0.05–0.07)	0.1 (0.08–0.11), <0.001	0.13 (0.12–0.15), <0.001
Spirituality and Religiosity	0.06 (0.05–0.08)	0.06 (0.05–0.08), 0.05	0.17 (0.16–0.19), <0.001
Stress and Resilience	0.08 (0.07–0.09)	0.14 (0.13–0.16), <0.001	0.19 (0.17–0.21), <0.001
**Hangzhou**	*Adj R^2^ (95% CI)*	*Adj R^2^ (95% CI), p-val.*	*Adj R^2^ (95% CI), p-val.*
WELL overall score	0.03 (0.02–0.03)	0.04 (0.03–0.04), <0.001	0.17 (0.16–0.19), <0.001
Experience of Emotions	0.05 (0.04–0.06)	0.05 (0.04–0.06), <0.001	0.07 (0.06–0.08), <0.001
Exploration and Creativity	0.04 (0.03–0.05)	0.04 (0.03–0.05), <0.001	0.16 (0.15–0.17), <0.001
Financial Sec. and Satisfaction	0.1 (0.09–0.12)	0.1 (0.09–0.12), 0.212	0.11 (0.1–0.13), <0.001
Physical Health	0.01 (0.01–0.02)	0.03 (0.02–0.04), <0.001	0.05 (0.04–0.06), <0.001
Purpose and Meaning	0.01 (0.01–0.01)	0.01 (0.01–0.02), <0.001	0.19 (0.18–0.21), <0.001
Sense of Self	0.05 (0.04–0.06)	0.06 (0.05–0.07), <0.001	0.11 (0.1–0.12), <0.001
Social Connectedness	0.04 (0.03–0.04)	0.04 (0.03–0.05), <0.001	0.06 (0.05–0.07), <0.001
Spirituality and Religiosity	0.02 (0.01–0.02)	0.02 (0.01–0.02), 0.312	0.03 (0.03–0.04), <0.001
Stress and Resilience	0.02 (0.01–0.02)	0.02 (0.02–0.03), <0.001	0.06 (0.05–0.07), <0.001
**New Taipei City**	*Adj R* ^2^ *(95% CI)*	*Adj R* ^2^ *(95% CI), p-val.*	*Adj R* ^2^ *(95% CI), p-val.*
WELL overall score	0.08 (0.06–0.1)	0.12 (0.1–0.14), <0.001	0.36 (0.33–0.38), <0.001
Experience of Emotions	0.08 (0.06–0.1)	0.1 (0.08–0.12), <0.001	0.15 (0.13–0.18), <0.001
Exploration and Creativity	0.03 (0.02–0.04)	0.05 (0.04–0.07), <0.001	0.26 (0.24–0.29), <0.001
Financial Sec. and Satisfaction	0.08 (0.06–0.1)	0.1 (0.08–0.12), <0.001	0.13 (0.11–0.16), <0.001
Physical Health	0.04 (0.02–0.05)	0.07 (0.05–0.09), <0.001	0.11 (0.09–0.13), <0.001
Purpose and Meaning	0.02 (0.01–0.03)	0.04 (0.02–0.05), <0.001	0.37 (0.35–0.4), <0.001
Sense of Self	0.06 (0.04–0.08)	0.07 (0.06–0.09), <0.001	0.17 (0.15–0.2), <0.001
Social Connectedness	0.05 (0.03–0.06)	0.06 (0.05–0.08), <0.001	0.12 (0.1–0.15), <0.001
Spirituality and Religiosity	0.03 (0.02–0.04)	0.03 (0.02–0.04), 0.826	0.07 (0.05–0.09), <0.001
Stress and Resilience	0.07 (0.05–0.09)	0.09 (0.07–0.11), <0.001	0.16 (0.14–0.19), <0.001

^1^ Different physical activity measures were used for the Hangzhou cohort than for the CABay Area and New Taipei City cohort.

## Data Availability

Individuals wishing to access the datasets used in this analysis should contact: Ann Hsing at annhsing@stanford.edu.

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
