# Peer review of "Contemplative Practices Behavior Is Positively Associated with Well-Being in Three Global Multi-Regional Stanford WELL for Life Cohorts"

_ijerph, 2022, doi:10.3390/ijerph192013485_

Round 1

Reviewer 1 Report

The authors describe a study that investigates the relationship of diverse  contemplative practice behaviors (CPB) and multiple dimensions of well-being since it is critical to have a more holistic consideration of these behaviors and extend them beyond what most studies have done with more uni-dimensional measures like mindfulness. The data presented describes data from three very diverse regions (i.e., San Francisco Bay Area, Hangzhou City and New Taipei City). In all three cohorts, the authors found statistically significant (p<0.05) positive associations between CPB and well-being, both overall and with all of the constituent domains of well-being, comparable to or stronger than the relationship with physical activity across most well-being outcomes. Thus, providing robust and cross-cultural evidence for a positive association between CPB and well being.

This is a very well performed study with great detail in description and statistical analysis. It is also timely and may serve to informe more detailed studies that are suggested in the Discussion.

One thought/suggestion that comes to mind is if the authors performed some sort of mediation analysis to see to what degree CPB mediates well being measures above and beyond physical activity.

Author Response

Thank you for your positive and constructive feedback. Please see the attaches file of our responses to your comments.

Reviewer 2 Report

Overall, I think this is an excellent study. I especially appreciate that, finally, researchers are looking at integrated, 'multiple' contemplative approaches. I also very much appreciate the cross-culture research design and implementation.

My only concerns, especially as someone who does work in the theory of contemplation and its practices, rest with the theory. As a general comment, I think the field of research into contemplative practices suffers from a lack of comprehensive and robust theory. With respect to this study, I think the authors might do a better job of defining contemplation and/or contemplative practice. Although I have enormous respect for Richie Davidson and his work, the definition he and Dahl have developed--"practical methods to bring about a state of enduring well-being or inner flourishing”--is far too vague and imprecise. One could easily fit a number of practices that are not contemplative under this conception, PA being but one. One line 149, you include yoga, mindfulness, and compassion cultivation as contemplative. But what is it about these practices that makes them contemplative? (And what do you mean by yoga? Hatha yoga, Ashtanga/Raja yoga ...?) I think some more elaboration on what constitutes contemplation and contemplative practice would improve the paper. This has significance to everything that follows with respect to any research in the field.

In lines 93-94, you write: "Contemplative practices emphasize self-awareness, self-regulation, and/or self-inquiry to enact a process of psychological transformation." Okay, but what about spiritual transformation (or well-being), and what are the boundaries or distinctions between psychological and spiritual transformation?

Might we need to do a better job of distinguishing between psychological, pro-social, and spiritual dimensions? Of course, all these are integrated within the individual, but my sense in reviewing the literature is that the spiritual dimension is not as well considered as it might be. 

Again, though, I appreciate your use of the four contemplative practices. We need more work that examines integrated approaches to contemplative practice. And might we draw guidance from the contemplative traditions themselves? The Buddhist Eightfold Path is an intentionally-designed integrated approach (with a strong body of theory), as is the Ashtanga/Raja yoga eight-limbed teaching; one could cite other examples of such integrated approaches from the contemplative traditions of east and west. Wilber and the Integral folks have used such approaches in designing contemporary approaches (e.g., "Integral Life Practice").  

I also appreciate your "Limitations" section. Yes, we have almost no longitudinal studies in the field, so causal relations remain unclear. One additional suggestion for future research is to include qualitative approaches to 'dig down' into the specifics of how contemplative practices contributed to well-being for various research subjects. 

Finally, a suspected typo on line 181. I presume 'form' is meant to be 'from.'

Author Response

Thank you for your positive and constructive feedback. Please find attached our response to your comments.
